# Transcriptional Profiling of Cardiac Cells Links Age-Dependent Changes in Acetyl-CoA Signaling to Chromatin Modifications

**DOI:** 10.3390/ijms22136987

**Published:** 2021-06-29

**Authors:** Justin Kurian, Veronica Bohl, Michael Behanan, Sadia Mohsin, Mohsin Khan

**Affiliations:** 1Center for Metabolic Disease Research (CMDR), LKSOM, Temple University, Philadelphia, PA 19140, USA; J.kurian7315@gmail.com (J.K.); veronicab298@gmail.com (V.B.); Michael.behanan@temple.edu (M.B.); 2Cardiovascular Research Center (CVRC), LKSOM, Temple University, Philadelphia, PA 19140, USA; sadia.mohsin@temple.edu; 3Department of Physiology, LKSOM, Temple University, Philadelphia, PA 19140, USA

**Keywords:** cardiac stem cells, metabolism, epigenetics, chromatin remodeling, acetyl-CoA, bioinformatics, RNA sequencing

## Abstract

Metabolism has emerged as a regulator of core stem cell properties such as proliferation, survival, self-renewal, and multilineage potential. Metabolites serve as secondary messengers, fine-tuning signaling pathways in response to microenvironment alterations. Studies show a role for central metabolite acetyl-CoA in the regulation of chromatin state through changes in histone acetylation. Nevertheless, metabolic regulators of chromatin remodeling in cardiac cells in response to increasing biological age remains unknown. Previously, we identified novel cardiac-derived stem-like cells (CTSCs) that exhibit increased functional properties in the neonatal heart (nCTSC). These cells are linked to a unique metabolism which is altered with CTSC aging (aCTSC). Here, we present an in-depth, RNA-sequencing-based (RNA-Seq) bioinformatic with cluster analysis that details a distinct epigenome present in nCTSCs but not in aCTSCs. Gene Ontology (GO) and pathway enrichment reveal biological processes, including metabolism, gene regulation enriched in nCTSCs, and STRING analysis that identifies a network of genes related to acetyl-CoA that can potentially influence chromatin remodeling. Additional validation by Western blot and qRT-PCR shows increased acetyl-CoA signaling and histone acetylation in nCTSCs compared to aCTSCs. In conclusion, our data reveal that the link between metabolism and histone acetylation in cardiac cells is altered with the aging of the cardiac tissue.

## 1. Introduction

Heart disease is characterized by lack of meaningful repair response by the injured cardiac tissue. Cardiomyocytes lost to injury are never replaced, which ultimately leads to adverse cardiac remodeling and heart failure. Cell-based therapies have offered promising alternatives to existing approaches to heart failure [1,2,3,4]. However, clinical results using several stem cell types have been largely underwhelming [5] and suggest that there is a need for a better understanding of the basic biology of stem cells and their response to injured myocardial environments. We have recently identified a novel cardiac-tissue-derived stem-like cell (CTSC) population in the heart which has a unique molecular and cellular profile compared to other known stem cells [6]. Interestingly, CTSCs possess a distinct metabolic phenotype that changes during the transition of the cells from neonatal to aged cardiac tissue concurrent with changes in proliferation and survival, suggesting that cellular metabolism is a mediator of CTSC function. Nevertheless, the precise role of metabolic signaling in regulating CTSC molecular and cellular function was not tested and remains unknown.

Metabolism has emerged recently as a mediator of stem cell fate as well as cellular and molecular function. Hallmark stem cell processes such as proliferation and survival are influenced by fluctuations in stem cell metabolism [7,8]. Various metabolites generated by stem cells during energy production act as secondary messengers, driving diverse signaling processes [9,10]. Studies have shown that certain metabolites such as acetyl coenzyme A (acetyl-CoA) continuously fluctuate, adjusting gene expression by modulating the epigenome in response to environmental changes [11,12]. Acetyl-CoA directly affects histone acetylation by transferring the acetyl group onto the histone lysine via histone acetyltransferases (HATs), thereby modulating epigenetic changes in gene expression [13,14]. In the cardiac context, epigenetic signaling pathways are known to change during embryonic, neonatal, and adult stages [15,16]. The existence of metabolic regulators of chromatin architecture in cardiac cells, however, is not well studied. Additionally, changes in the metabolic profile of cardiac cells during the transition from young to aged and the role of the acetyl-CoA signaling pathway in regulating chromatin modification remain unknown.

In this article, CTSCs were isolated from neonatal (2-day-old; nCTSCs) and aged (2-year-old; aCTSC) hearts and subjected to bulk RNA-Seq to identify metabolic signaling pathways related to chromatin modulation. Comparative analysis of neonatal and aged CTSCs shows unique metabolic and chromatin modifying processes, and reveals acetyl-CoA regulators and signaling pathways being altered with CTSC age in the heart. These findings provide a novel insight connecting cellular metabolism with the maintenance of chromatin architecture in cardiac cells during transition from young to aged and thereby provide a new dimension to functional regulation in CTSCs.

## 2. Results

### 2.1. Distinct CTSC Profiles in Response to Aging as Revealed by Transcriptomic Analysis

In this study, we describe the identification of novel metabolic modulators of chromatin architecture in cardiac-derived stem-like cells (CTSC), a novel cell type in the heart, through bioinformatic and molecular approaches. For this study, we isolated two distinct CTSC populations—one from 2-day-old neonatal (nCTSC) cardiac tissue and one from aged, 2-year-old (aCTSC) cardiac tissue as previously described. This was followed by bulk RNA sequencing and detailed bioinformatic-based analyses to understand the transcriptomic profile of each CTSC line (Figure 1A). The raw count files were retrieved after sequencing, formatted, and uploaded onto the RStudio platform for gene expression analysis via DESeq2. Direct comparison of nCTSC with aCTSC identified a total of 14,274 genes, of which 31% were significantly upregulated in nCTSC and 30% were significantly downregulated (Figure 1B). Scatter plot (Figure 1C) and heatmap (Figure 1D) representations visualize the up- and downregulation of genes between nCTSC and aCTSC. 

Next, we assessed transcriptomic differences between nCTSCs and aCTSCs by performing a principal coordinates assay (PCoA) that visualized dissimilarity between the two samples (Figure 2A). In our data, the PCoA plot not only shows that the CTSC replicates from neonatal and aged hearts are low in variance, but also that there are significant differences between nCTSC replicates and the aCTSC transcriptional profile. These changes may detail the developmental transition that occurs as neonatal CTSCs eventually age and acquire aCTSC phenotypic characteristics along with the aging of the cardiac tissue. Finally, the Pearson correlation plot (Figure 2B) shows that even though the nCTSCs and aCTSCs are distinct subpopulations, the overall correlation between the two CTSC populations is high because they both originated in the heart and were isolated under similar conditions. Collectively, data analysis reveals that CTSC transcriptome alters with the age of the cardiac tissue, and this affects the neonatal cells more than the aged CTSCs.

### 2.2. Metabolic Processes Are Upregulated in nCTSC as Shown by Clustering Analysis

The next set of analyses focused on discovering functionalities occurring between the two CTSC populations and identifying biological relevance. For this, we initially clustered differentially expressed genes (DEG) through k-means methodology and found five clusters identified by the elbow method to be ideal for subsequent analyses (Figure 2C). Genes within each cluster were determined through comparison and identification of the minimum distance to cluster mean. Once generated, the clusters were subjected to downstream functional enrichment analyses, such as Gene Ontology (GO) and Ingenuity Pathway Analysis (IPA). GO analysis of biological processes showed significant GO terms unique to each cluster. For example, cluster 1 consists of GO terms such as Cellular Lipid Metabolic Process (FDR 1.52 × 10^−3^), Monocarboxylic Acid Metabolic Process (FDR 3.8 × 10^−3^), and additional metabolic GO terms, indicating that this cluster is responsible primarily for metabolism (Figure 3A). GO analysis of clusters 2, 3, 4, and 5 shows them to be responsible for cellular processes of immune response (Figure 3B), development (Figure 3C), gene expression (Figure 3D), and cell cycle (Figure 3E), respectively (Appendix A). GO analysis confirmed that the canonical pathway was enriched, as determined by IPA, to similar biological terms and signaling pathways for each cluster. Interestingly, in cluster 4, GO terms involved regulation of gene expression and showed significant enrichment of processes such as ‘Embryonic Stem Cell Differentiation into Cardiac Lineages’ and ‘Mouse Embryonic Stem Cell Pluripotency’ in the nCTSCs. Moreover, pathway analysis of cluster 4 revealed enriched terms that may suggest a unique chromatin remodeling, possibly related to maintaining the stem-like phenotype in nCTSCs. This draws a parallel with previous studies that focus on how chromatin remodeling is essential to stem cell fate in either maintaining pluripotency or progressing to differentiation. Altogether, functional analysis reveals heterogenous clusters of genes involved in many essential activities, such as metabolism and regulation of gene expression. 

### 2.3. ACSS2 Is Highly Enriched in nCTSCs Compared to aCTSCs

We previously reported that in vitro characterization of CTSCs showed a significant metabolic shift occurring as the hearts age [6]. The neonatal heart is largely dependent on glycolytic metabolism, while oxidative phosphorylation is predominant in aged hearts. To identify metabolic regulators connected with changes in gene expression, we conducted further analysis of cluster 1 by REVIGO analysis that removed redundant GO terms and defined prominent terms such as monocarboxylic acid metabolism significantly enriched in nCTSC compared to aCTSC (FDR 3.8 × 10^−3^) (Figure 4A). A heatmap was generated to visualize the genes present in that GO term (Figure 4B). Among those genes significantly upregulated in nCTSCs compared to aCTSCs was Acetyl coenzyme A synthetase short-chain family member 2 (ACSS2). ACSS2 is an essential cytosolic protein involved in lipid metabolism and generates Acetyl-CoA. Previous studies show that ACSS2 is involved in many functions beyond metabolism, such as chromatin regulation. To identify potential ACSS2 targeting genes in nCTSCs and to understand their biological relevance, a table was generated showing the most significantly altered genes present in each of the top 10 GO terms within all clusters (Table 1). Since cluster 1 represented DEGs that belonged to metabolism, the top identified genes in each GO term within cluster 1 were subjected to STRING database and visualized through 

Cytoscape to predict the interaction between top identified genes in each GO term and ACSS2 (Appendix A). The results showed unique biological signaling pathways regulated by the top identified genes in each GO term (Appendix A). Interestingly, out of all the other top identified genes per GO term in cluster 1, only Sqle was predicted to directly target ACSS2 in nCTSCs (Figure 4C). Finally, we investigated additional genes regulated by ACSS2 in nCTSCs within our data. For this purpose, we utilized a published STRING database for the top 50 genes that are predicted to interact with ACSS2 (Appendix A) and compared them to our list of DEGs altered in nCTSCs compared to aCTSCs. This analysis identified 21 significantly altered genes that directly interact with ACSS2 within the entire group of DEGs altered in cluster 1 (Figure 4D). Our network reveals many metabolically upregulated terms, such as Aox3 and Aldh7a1, but it also includes other genes, such as PCCA and Sod2, that are known to possess histone regulatory functions. Collectively, our data identify ACSS2 as a central nodal signaling pathway in nCTSC, and one that is predicted to be regulated by several metabolic gene targets with potential to affect chromatin regulation as well. 

### 2.4. Altered ACSS2 Signaling Pathway and Chromatin Modifications in nCTSCs

Metabolites are shown to be involved in the regulation of gene expression [10,12]. Epigenetic alterations occur through the interaction of metabolites on histone residues to either open or condense chromatin. Transcriptional access to specific parts of the chromatin leads to phenotypical changes within the cells. Our data identified metabolic regulators of acetyl-CoA in nCTSCs, and we hypothesized that increased acetyl-CoA may be linked to regulation of chromatin modification in nCTSCs and is altered with CTSC age. To confirm the hypothesis, we conducted additional analysis and identified “Chromatin Remodeling” as a top enriched term (FDR 2.57 × 10^−4^) in cluster 4 GO analysis, along with many upregulated genes present in the nCTSC population (Figure 5A). Western blot analysis confirmed our RNA-seq data and showed a significantly increased acetylated Histone H3 to total Histone H3 ratio in the nCTSCs compared to the aCTSCs (Figure 5B). Additionally, significant increases in expression of ATP-citrate lyase (ACLY), the primary enzyme involved in the cytosolic synthesis of acetyl-CoA, and cytoplasmic acetyl-CoA synthetase (AceCS1) were observed in nCTSCs compared to aCTSCs (Figure 5C). In concordance, qRT-PCR analysis confirmed that upregulation of ACLY and ACSS2 is associated with cytoplasmic acetyl-CoA synthesis, along with low expression of mitochondrial protein CPT1a (0.6193FC) and high expression of cytosolic PDHB (2.725FC), suggesting the involvement of glycolytic flux in driving synthesis of acetyl-CoA in nCTSCs compared to aCTSCs (Figure 5D). Altogether, our data suggest that chromatin remodeling in nCTSCs may be altered by the metabolic regulation of ACSS2 by synthesizing more cytoplasmic acetyl-CoA, which potentially acetylates histone.

## 3. Discussion

Our findings here identify metabolic modifiers of chromatin modulation in a novel stem-like cell population termed as CTSCs in the heart during transition from young to old age. Additionally, our data identify acetyl-CoA signaling as a central hub for upstream metabolic regulators and downstream alteration of chromatin architecture. CTSCs from the neonatal heart demonstrate increased activation of the cytosolic acetyl-CoA signaling pathway together with chromatin modifications, in comparison to aCTSCs.

Cell-based therapeutics have emerged recently as a viable strategy for cardiac repair following myocardial injury. Several stem cell types have been used, but their clinical applications [3,5] were largely underwhelming due to massive loss of donated stem cells in the ischemic cardiac environment [17,18]. There is a greater need for understanding the basic biology of stem cells before clinical testing. We have recently identified a novel cell population in the heart with stem cell-like properties termed cardiac-derived stem-like cells (CTSCs) [6]. Characterization of CTSC population isolated from 2-day-old (nCTSCs) and 2-year-old hearts (aCTSC) shows a unique transcriptomic signature of cells in comparison to other known stem cell types, including cardiac tissue-derived stem/progenitor cells and cardiac cells (cardiomyocytes, fibroblasts, endothelial cells), and also shows enhanced functional properties during the neonatal stages that are lost with CTSC age. In concurrence with our findings, cardiac stem cells from the neonatal stages are known to exhibit strong regenerative properties that are altered with age [19,20]. Additionally, Castaldi and colleagues found that aged cardiac progenitor cells exhibit senescent properties, impaired mitochondrial respiration, and decreased ATP generation [20]. Whether metabolic signaling pathways play a central role in maintaining cellular function and gene expression in cardiac-derived stem cells remains unknown. 

Cellular metabolism has emerged recently as a regulator of stem cell fate as well as molecular and cellular function [21,22]. Changes in environmental conditions alter energy generation in stem cells leading to a shift from a quiescent to proliferative or differentiated state [23]. In the cardiac context, cardiac stem/progenitor cells reside in hypoxic niches in a quiescent state while operating under glycolysis [24]. Emergence from the niche is associated with a shift in metabolism to oxidative phosphorylation, concurrent with loss of quiescence and acquisition of a differentiated state. Interestingly, nCTSCs exhibit glycolytic metabolism and increased proliferation rates that are largely altered with age [6]. Analysis of molecular signaling showed nCTSCs as expressing stem cell and pluripotent markers, as well as enhanced expression of cell cycle regulators. Whether cellular metabolism or metabolic regulators were responsible for unique CTSC transcriptomic profile, including ability to modulate gene expression and their alteration with age, was not tested. Therefore, in this study, we conducted data mining using bulk RNA-sequencing data from nCTSCs and aCTSCs to identify metabolic regulators that are altered with age. Our results show that the nCTSC transcriptome is significantly different from that of aCTSCs, thus implying susceptibility of CTSCs to age-associated changes in cardiac environment. These results are in concurrence with studies showing age-associated changes in cardiac stem/progenitor cells, attenuating stem cell properties and functionality [20,25]. Bioinformatic analyses revealed that nCTSC transcriptome was divided into five clusters, with cellular metabolism as one of the most prominent categories that is significantly altered in nCTSC compared to aCTSCs. Additionally, analyses identified several key metabolic genes related to the monocarboxylic acid metabolic signaling pathway being significantly altered in nCTSCs compared to aCTSCs, suggesting a critical role is played by cellular metabolism in regulation of nCTSC function, and that it is altered with age. Our results are in accordance with some recent studies that implicate a critical role of metabolism as a determinant of cell therapy effectiveness [26,27,28].

The next question that we addressed through bioinformatic approaches was how the metabolic regulators in nCTSCs affect gene expression. For this purpose, we hypothesized that metabolic regulators influence levels of metabolites and acetyl-CoA to alter gene expression. Various metabolites are known to act as secondary messengers, altering gene expression and signaling pathways in response to environmental changes. Nevertheless, the mechanistic basis for how metabolites influence signaling pathways is only now being uncovered and carries implications for regulation of molecular and cellular function in different cell types. The central metabolite, acetyl-CoA, and its relationship with histone acetylation was first reported in yeast and mammalian cells and is now well established [11,12]. Recent evidence indicates that Acetyl-CoA is essential for stem cell function and maintenance of pluripotency [23]. Pluripotent stem cells (PSCs) exhibit increased proliferative rates, supported primarily by glycolysis and shunting of glycolytic intermediates for acetyl-CoA generation [29]. Additionally, Moussaieff et al. found that PSCs produce cytosolic acetyl-CoA through glycolysis and the pyruvate-derived citrate flux via ATP citrate lyase (ACLY), while PSC differentiation shuts down this mechanism [7]. Increased cytosolic acetyl-CoA levels in PSC regulate histone acetylation, which maintains open euchromatin state, pluripotency, and self-renewal in PSCs [7]. In the cardiac context, acetyl-CoA plays a central role in cardiac energy generation, including allosteric modulation of fatty-acid, glucose oxidation, and acetylation control of cardiac energy pathways that are altered with cardiac disease states [30,31]. Nevertheless, the metabolic pathways connected to acetyl-CoA and chromatin modulation in cardiac-derived stem cells remain unknown. Our findings show that age-dependent alterations in metabolic pathways and acetyl-CoA levels in CTSCs derived from the heart, as well as changes in histone acetylation and chromatin modifications, are in concurrence with recent findings on acetyl-CoA mediated chromatin regulation in other organ systems. In summary, our findings link metabolism to maintenance of chromatin state and gene regulation in CTSCs, which are largely altered with age. 

## 4. Materials and Methods

### 4.1. Cell Isolation and Culture

Cell isolation and maintenance of CTSCs were performed as previously described [6]. Additional details in the online Appendix A.

### 4.2. RNA Sequencing

CTSCs were prepared for bulk RNA sequencing with three replicates of each CTSC line, consisting of 1 million cells per replicate cultured at passage 10 for 48 h prior to sequencing. Samples were sent out to GeneWiz for RNA isolation, cDNA library construction, and sequencing. In brief, RNA was isolated from cells by GeneWiz, including mRNA sequencing via PolyA selection. Library preparation involved fragmentation and random priming, strand cDNA synthesis, end repair, and adapter ligation with polymerase chain reaction (PCR) enrichment for sequencing. Sequencing was carried out on Illumina HiSeq System with 2 × 150 bp PE HO configuration.

### 4.3. Bioinformatics Analysis

Raw data quality was evaluated with FastQC. Reads were trimmed using Trimmomatric v. 0.36 to remove adapter sequences and poor-quality nucleotides. Reads were then mapped onto Mus musculus GRCm38 reference genome using STAR aligner v. 2.5.2b. Genome was available on ENSEMBL. Generated BAM files were used to extract unique gene-hit counts using FeatureCounts from Subread package v. 1.5.2. Differentially expressed genes (DEG) were identified on mapped reads, followed by downstream differential expression analysis using R package DESeq2. Genes with less than 5 reads per sample were removed. The Wald test was used to statistically identify genes with *p* value < 0.05. Using regularized logarithm (rlog) function, count data were transformed for visualization on a log_2_ scale. K-means clustering was performed to identify subgroups of genes that were subjected for downstream enrichment analysis. Number of clusters were determined using elbow method. 

### 4.4. Functional Analysis

Clustered DEGs were uploaded onto the gene ontology enrichment analysis and visualization (GOrilla) server to identify significantly enriched Biological Process GO terms. Heatmaps were generated using R package ggplots on specific GO terms. Clustered genes were uploaded for pathway enrichment through Ingenuity Pathway Analysis (IPA) software to identify canonical pathways for biological relevance. Cytoscape’s STRING database was used to identify and generate functional protein interaction networks. The GO term list generated from each cluster via GOrilla was uploaded onto online server Reduce + Visualize Gen Ontology (REViGO) to summarize and remove redundant terms. 

### 4.5. Reverse Transcriptase Polymerase Chain Reaction

CTSCs RNA extraction and purification were completed using manufacturer’s instructions (Qiagen, Germantown, MD, USA). After RNA quantification (Nanodrop), cDNA was created using iScript cDNA Synthesis Kit (Bio-Rad, Hercules, CA, USA). Amplification was conducted using SimpliAmp thermocycler (Applied Biosystems, Waltham, MA, USA). Real-time polymerase chain reaction (RT-PCR) was conducted by running samples on StepOnePlus RT-PCR System (Applied Biosystems). Primers were created from Life Technologies, and the sequences are included in the Appendix A.

### 4.6. Western Blot

Immunoblot analysis was performed as previously described [6,32,33] and list of antibodies is included in Appendix A.

### 4.7. Statistical Analysis

Statistical analysis was performed using unpaired Student’s *t* test for data comparing two groups. For data that do not exhibit normal distribution, the Mann–Whitney test was used. All data sets were assessed for normality using the Shapiro–Wilk test. *p* < 0.05 is considered statistically significant. Error bars represent ±SD. Statistical analysis is performed using Graph Pad prism v 8.0 software (GraphPad Software Inc., San Diego, CA, USA).

## 5. Conclusions

Here, we report on the interplay between metabolism and chromatin modifications present in nCTSCs. Our previous study reports that the nCTSCs maintain stemness, are more proliferative with a unique secretome, and are more glycolytically driven than their aged counterparts. RNA-seq analysis reveals that the increased Acyl-coenzyme A synthetase short-chain family member 2 (ACSS2) levels lead to more Acetyl-CoA availability for histone acetylation. This regulated gene expression through metabolism may be a promising avenue for future studies focusing on how stem cells maintain their therapeutic profile. 

## Figures and Tables

**Figure 1 ijms-22-06987-f001:**
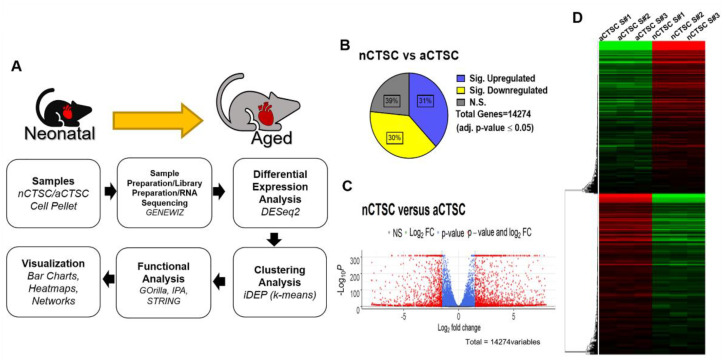
Schematic representation of workflow. (**A**) Cells from distinctly aged mice were isolated and cultured. Cell pellets from 3 replicates were used for RNA extraction and sequencing. Downstream analysis was performed by comparing nCTSC vs. aCTSC genes, with adjusted *p*-values less than 0.05. (**B**) Direct comparison of differentially expressed genes (DEG) that are significantly upregulated and downregulated between nCTSC and aCTSC. Total of 14,274 genes were identified. (**C**) Volcano plot visualizing genes based on log_2_ Fold-Change vs. −log_10_ (adjusted *p*-value). Thresholds of log_2_ FC (±1.5) and −log_10_ (adjusted *p* value) (>5). Each dot represents a gene and is colored in respect to its value. (**D**) Heatmap was generated using the rlog values (DESeq2) of the top 10,000 most-variable genes present in transcriptomic comparison. Values are normalized in row z-score range between −2 and 2.

**Figure 2 ijms-22-06987-f002:**
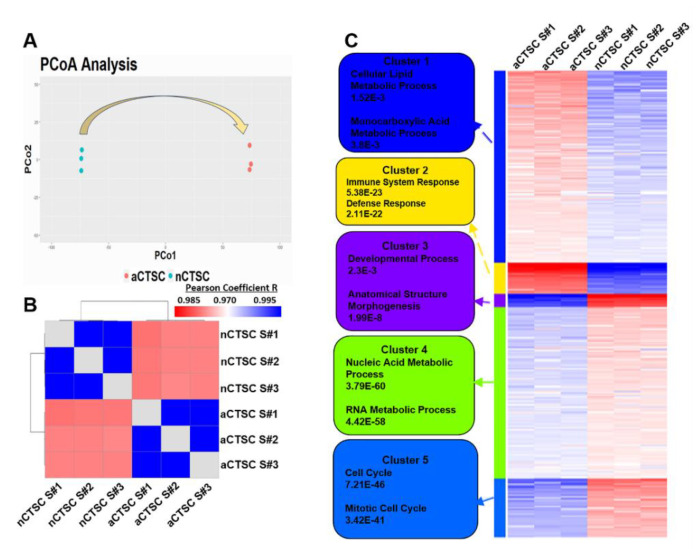
Distinct processes present in each cluster through Gene Ontology (GO) enrichment. (**A**) Principal coordinates analysis of transcriptomic data comparing the two samples. (**B**) Correlation matrix compares nCTSC with aCTSC using Pearson correlation coefficient. (**C**) Heatmap visualizing five clusters distinctly identified using elbow method and subjected to GO enrichment.

**Figure 3 ijms-22-06987-f003:**
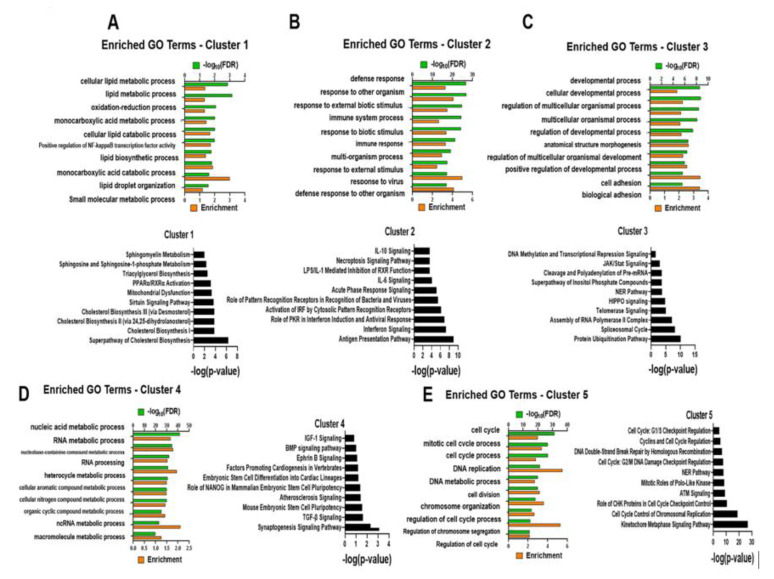
Functional analysis of each cluster. Top 10 GO terms generated from GO Biological Processes analysis for each cluster depicting −log_10_ (FDR) and enrichment values and Ingenuity Pathway Analysis (IPA) together with Top 10 canonical pathways presented based on –log(*p*-value) for (**A**) Cluster 1, (**B**) Cluster 2, (**C**) Cluster 3, (**D**) Cluster 4, (**E**) Cluster 5.

**Figure 4 ijms-22-06987-f004:**
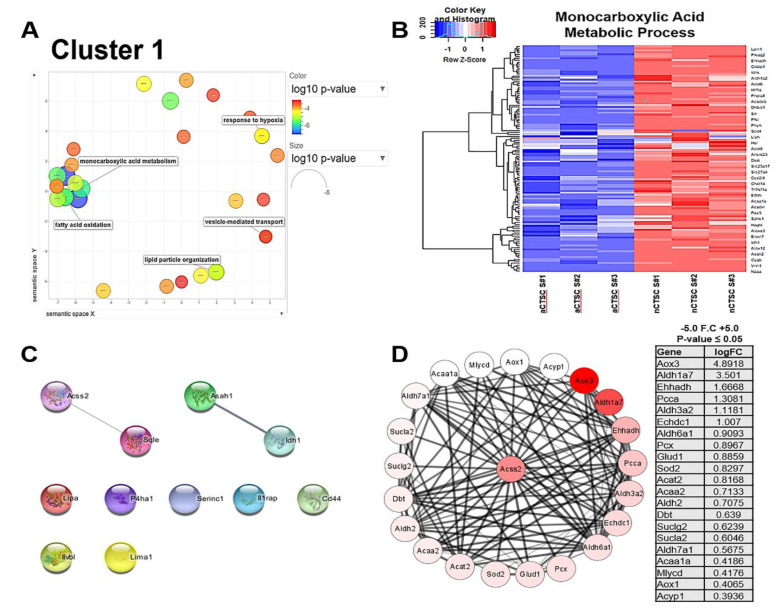
Metabolic processes in cluster 1 related to ACSS2. (**A**) REVIGO analysis of cluster 1 visualizing similar GO terms grouped together in semantic space. (**B**) Heatmap of GO terms: monocarboxylic acid metabolic process. (**C**) Top 10 genes present in cluster 1 identified by STRING database and visualized through Cytoscape to show potential interaction with ACSS2. (**D**) Protein–protein interaction network of genes from cluster 1 that directly interact with Acyl-CoA synthetase short-chain family member 2 protein (ACSS2). Color of each protein ranges from white to red and corresponds to each gene logFC.

**Figure 5 ijms-22-06987-f005:**
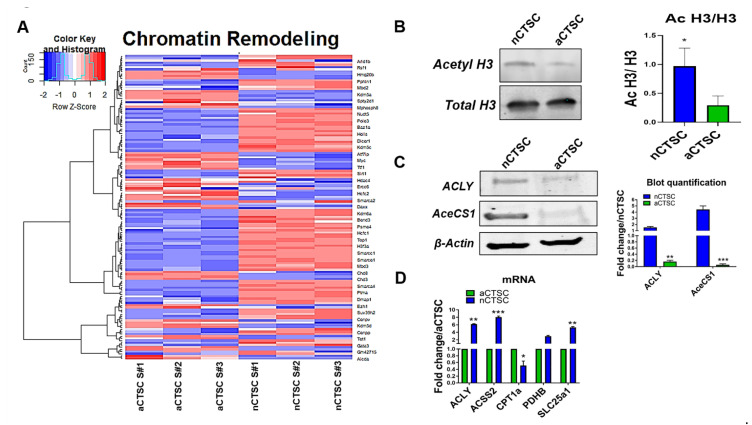
Acetyl-CoA-mediated modulation of chromatin modifications and histone acetylation in nCTSC. (**A**) Assessment of genes involved in GO term: chromatin remodeling. (**B**) Increased expression of Acetyl H3: total H3 found in nCTSC compared to aCTSC (n = 3). (**C**) Immunoblot revealing increased protein expression of ACLY and AceCS1 in the nCTSC (n = 3). (**D**) Increased mRNA expression of ACSS2 and glycolytic enzymes along with reduced fatty-acid metabolic activity present in nCTSC by qRT-PCR (n = 3). nCTSC vs. aCTSC * *p*  < 0.05, ** *p*  < 0.01, *** *p*  <  0.001.

**Table 1 ijms-22-06987-t001:** Top 10 genes/GO terms in each cluster.

Cluster 1	Metabolism	GO Description	Top Gene	LogFC	adj. *p*-Value
	GO:0044255	Cellular lipid Metabolic process	**Lipa**	1.880428797	8.3778 × 10^−265^
	GO:0006629	Lipid metabolic process	**Lima1**	1.386559217	4.9641 × 10^−287^
	GO:0055114	oxidation-reduction process	**P4ha1**	1.216435527	4.0584 × 10^−292^
	GO:0032787	monocarboxylic acid metabolic process	**Idh1**	1.584898068	1.4915 × 10^−241^
	GO:0044242	cellular lipid catabolic process	**Asah1**	0.963340936	1.8342 × 10^−167^
	GO:0051092	positive regulation of NF-kappaB transcription factor activity	**Il1rap**	2.253245162	4.1379 × 10^−303^
	GO:0008610	lipid biosynthetic process	**Serinc1**	0.986106666	7.161 × 10^−242^
	GO:0008610	lipid biosynthetic process	**Serinc1**	0.986106666	7.161 × 10^−242^
	GO:0072329	monocarboxylic acid catabolic process	**Ilvbl**	2.185676922	1.048 × 10^−151^
	GO:0034389	lipid droplet organization	**Sqle**	0.920963676	4.2693 × 10^−134^
	GO:0044281	small molecule metabolic process	**Cd44**	0.996198493	5.32 × 10^−285^
**Cluster 2**	**Immune**	**GO Description**	**Top Gene**	**logFC**	**adj. *p*-Value**
	GO:0006952	defense response	**Rtp4**	4.132592871	1.8172 × 10^−298^
	GO:0051707	response to other organism	**Usp18**	6.2573426	1.7065 × 10^−301^
	GO:0043207	response to external biotic stimulus	**Stat1**	2.589408409	1.0021 × 10^−269^
	GO:0002376	immune system process	**Ifi44**	6.77996421	8.9746 × 10^−266^
	GO:0009607	response to biotic stimulus	**Mlkl**	4.440672777	7.6562 × 10^−261^
	GO:0006955	immune response	**Cyba**	6.401161196	2.3696 × 10^−247^
	GO:0051704	multi-organism process	**Ifit2**	2.868875789	7.9652 × 10^−245^
	GO:0009605	response to external stimulus	**Gbp3**	4.860447879	4.3027 × 10^−236^
	GO:0009615	response to virus	**Oasl2**	7.648377266	2.6227 × 10^−227^
	GO:0098542	defense response to other organism	**Irgm2**	4.930420913	4.1127 × 10^−205^
**Cluster 3**	**Development**	**GO Description**	**Top Gene**	**logFC**	**adj. *p*-Value**
	GO:0032502	developmental process	**Slc6a17**	−3.493587034	1.5326 × 10^−295^
	GO:0048869	cellular developmental process	**Kazald1**	−4.33489428	1.468 × 10^−281^
	GO:0051239	regulation of multicellular organismal process	**Fam19a5**	−4.123121971	4.1061 × 10^−272^
	GO:0032501	multicellular organismal process	**Plxnb1**	−4.447505569	3.2518 × 10^−270^
	GO:0050793	regulation of developmental process	**Tenm3**	−7.891281718	2.4011 × 10^−266^
	GO:0009653	anatomical structure morphogenesis	**Jag1**	−3.608377716	3.508 × 10^−228^
	GO:2000026	regulation of multicellular organismal development	**Prdm16**	−3.568565321	9.015 × 10^−253^
	GO:0051094	positive regulation of developmental process	**P2rx7**	−6.008024776	1.4826 × 10^−187^
	GO:0007155	cell adhesion	**Sele**	−5.033733739	4.4538 × 10^−253^
	GO:0022610	biological adhesion	**Col12a1**	−3.822295006	9.6969 × 10^−249^
**Cluster 4**	**Gene Expression**	**GO Description**	**Top Gene**	**logFC**	**adj. *p*-Value**
	GO:0090304	nucleic acid metabolic process	**Set**	−1.108981655	1.9585 × 10^−289^
	GO:0016070	RNA metabolic process	**Ddx3x**	−1.081698642	3.464 × 10^−268^
	GO:0016070	RNA metabolic process	**Ddx3x**	−1.081698642	3.464 × 10^−268^
	GO:0006139	nucleobase-containing compound metabolic process	**Ran**	−1.018798075	1.1 × 10^−244^
	GO:0006396	RNA processing	**Tardbp**	−1.265502477	5.7847 × 10^−261^
	GO:0046483	heterocycle metabolic process	**Dhx9**	−1.062768239	1.0373 × 10^−192^
	GO:0006725	cellular aromatic compound metabolic process	**Gspt1**	−1.02188382	1.6288 × 10^−189^
	GO:0034641	cellular nitrogen compound metabolic process	**Mbnl1**	−0.97759175	6.1882 × 10^−188^
	GO:1901360	organic cyclic compound metabolic process	**Smc1a**	−1.255023232	1.3984 × 10^−187^
	GO:0034660	ncRNA metabolic process	**Rpl11**	−0.834339298	5.4663 × 10^−171^
	GO:0043170	macromolecule metabolic process	**Srsf3**	−1.232300645	7.5139 × 10^−239^
**Cluster 5**	**Cell Cycle**	**GO Description**	**Top Gene**	**logFC**	**adj. *p*-Value**
	GO:0007049	cell cycle	**Birc5**	−2.60656371	2.1503 × 10^−303^
	GO:1903047	mitotic cell cycle process	**Tgfb1**	−1.695288695	5.3792 × 10^−306^
	GO:0022402	cell cycle process	**Prc1**	−1.763009143	7.0517 × 10^−300^
	GO:0006260	DNA replication	**Ccdc88a**	−1.596474351	1.4062 × 10^−302^
	GO:0006259	DNA metabolic process	**Nasp**	−2.48862378	1.2227 × 10^−294^
	GO:0051301	cell division	**Cad**	−1.549902684	3.5803 × 10^−285^
	GO:0051276	chromosome organization	**Cenpe**	−1.775921663	6.4099 × 10^−272^
	GO:0010564	regulation of cell cycle process	**Ranbp1**	−0.391960617	1.1679 × 10^−295^
	GO:0051983	regulation of chromosome segregation	**Racgap1**	−1.68593893	8.6581 × 10^−262^
	GO:0051726	regulation of cell cycle	**Cenpf**	−2.099842197	1.7343 × 10^−268^

## Data Availability

Data is contained within the article and Appendix A.

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
