# Peer review of "Transcriptional Profiling of Cardiac Cells Links Age-Dependent Changes in Acetyl-CoA Signaling to Chromatin Modifications"

_ijms, 2021, doi:10.3390/ijms22136987_

Round 1

Reviewer 1 Report

This is a nice study that performed a comparative transcriptional profiling of the newly identified murine cardiac derived stem like cells (CTSC) at two developmental stages (neonate and adult). In this paper, the authors used RNA Sequencing approach followed by in depth bioinformatic study that allowed them to establish an age dependent correlation between changes in acetyl-coA signaling and histone acetylation in cardiac cells. The manuscript is well written and structured. However, the only concern I have is that it would be interesting if the authors have included in their comparative transcriptional profiling, data from other endogenous cardiac stem cell types (external control) and other developmental stages.

Minor comments:

-Please improve the resolution of Figure 3

Line 13: "the regulation....."

Line 141: "...as determined by..."

Line 282: "...that we addressed is...."

Line 283: "through bioinformatic approaches" it would be better if this goes earlier after "The next question that we addressed"

All the best.

Reviewer 2 Report

This manuscript represents a development of a previous study by Khan et al. Here, the authors aimed to compare the gene profiles from neonatal- and aged- cardiac derived stem like cells (CTSCs). Using bioinformatics with cluster analysis, the epigenome present in neonatal and aged CTSCs was detailed and revealed peculiar differences in markers of biological processes including metabolism and gene regulation between the two cell populations.

It is not surprising that stemness properties resulted increased in stem-like cells derived from younger subjects in comparison to cells from older ones.

Nevertheless, in this study the authors focused on cell metabolism markers influencing chromatin remodeling, and found this pattern increased in neonatal CTSCs compared to aged ones. These findings paved the way to define some mechanisms in cardiac aging.

In my opinion, the manuscript is suitable to be published after the revision of typing errors and English style. Furthermore, the Discussion could be improved by reorganizing this section underlining the conclusion remarks of the study, as in this form the Discussion appeared without an efficacious ending.
